# The CRK5 and WRKY53 Are Conditional Regulators of Senescence and Stomatal Conductance in *Arabidopsis*

**DOI:** 10.3390/cells11223558

**Published:** 2022-11-10

**Authors:** Paweł Burdiak, Jakub Mielecki, Piotr Gawroński, Stanisław Karpiński

**Affiliations:** Department of Plant Genetics, Breeding, and Biotechnology, Institute of Biology, Warsaw University of Life Sciences, 02-776 Warsaw, Poland

**Keywords:** senescence, receptor-like kinase, transcription factor, water-use efficiency, stomatal conductance, foliar temperature, development, osmotic stress

## Abstract

In *Arabidopsis thaliana*, cysteine-rich receptor-like kinases (CRKs) constitute a large group of membrane-localized proteins which perceive external stimuli and transduce the signal into the cell. Previous reports based on their loss-of-function phenotypes and expression profile support their role in many developmental and stress-responsive pathways. Our study revealed that one member of this family, CRK5, acts as a negative regulator of leaf aging. Enrichment of the *CRK5* promoter region in W-box *cis*-elements demonstrated that WRKY transcription factors control it. We observed significantly enhanced *WRKY53* expression in *crk5* and reversion of its early-senescence phenotype in the *crk5 wrky53* line, suggesting a negative feedback loop between these proteins antagonistically regulating chlorophyll *a* and *b* contents. Yeast-two hybrid assay showed further that CRK5 interacts with several proteins involved in response to water deprivation or calcium signaling, while gas exchange analysis revealed a positive effect of CRK5 on water use efficiency. Consistent with that, the *crk5* plants showed disturbed foliar temperature, stomatal conductance, transpiration, and increased susceptibility to osmotic stress. These traits were fully or partially reverted to wild-type phenotype in *crk5 wrky53* double mutant. Obtained results suggest that WRKY53 and CRK5 are antagonistic regulators of chlorophyll synthesis/degradation, senescence, and stomatal conductance.

## 1. Introduction

Senescence is essential for plant fitness since it controbutes to the redistribution of micro- and macro-nutrients to growing and reproductive organs [1]. However, premature senescence, induced by stress, leads to reduced yield and quality of crops [2]. This process is preceded by molecular changes regulated by almost all plant hormones and many small signaling molecules, e.g., intracellular Ca^2+^ or reactive oxygen species (ROS) [3,4]. The most visible symptoms of plant senescence are the degradation of chlorophyll, proteins, lipids, and RNA, which results in chloroplast disassembly. In *Arabidopsis*, 12–16% of all genes are down- or upregulated during leaf aging, indicating a high activity of transcription factors (TFs) during this developmental stage [5]. Many WRKY proteins are activated during senescence [6]. Previous reports showed that WRKY54 and WRKY70 coperate as negative, whereas WRKY6, WRKY22, WRKY30, and WRKY53 act as positive regulators of senescence [7,8,9,10,11]. Out of 75 WRKY TFs present in the *Arabidopsis* genome, especially WRKY53 has been described in more detail. Transcriptomic data showed that *WRKY53* is positively regulated by salicylic acid (SA) and hydrogen peroxide, while jasmonic acid (JA) has a negative effect on its expression [12,13,14]. This TF acts as a positive regulator of senescence, regulating catalases and many genes involved in transport and remobilization processes [8,9,10]. Interestingly, apart from its function in plant senescence, WRKY53 has also been found to play a positive role in regulating plant sensitivity to salt stress [15].

The majority of internal and external stimuli in plants are perceived by receptor-like kinases (RLKs). They are composed of an extracellular domain, a single transmembrane region, and a highly conserved cytoplasmic domain possessing serine/threonine kinase activity. The extracellular domain of RLKs shows great diversity and can sense a variety of extracellular ligands, making the basis for their classification into subfamilies [16]. Recently, more and more research has been focused on unraveling the functions of cysteine-rich receptor-like kinases (CRKs), which constitute one of the largest subgroups of RLKs with 44 members in *Arabidopsis thaliana*. They possess an extracellular domain harboring two conserved cysteine-rich motifs (C-8X-C-2X-C), which might act as sensors during ROS and redox signaling and lead to a conformational change, for example, through the opening of disulfide bridges [17]. Based on their expression profile in response to SA, chloroplast retrograde signaling, pathogen attack [18], and loss-of-function phenotypes [17], CRKs are promising regulatory proteins for abiotic and biotic stress-responsive pathways. Several reports revealed the function of some CRK members, e.g., showing that CRK2 increases salt tolerance by regulating callose deposition [19], CRK6 and CRK7 play a protective role specifically in apoplastic oxidative stress [20], while CRK36 and CRK45 regulate abscisic acid (ABA)-mediated signaling pathways in response to osmotic stress [21,22]. Previous reports also showed that constitutive expression of *CRK5* correlated with enhanced leaf growth and enhanced pathogen resistance, while a steroid-inducible expression resulted in the activation of hypersensitive response (HR)-like cell death upon pathogen infection [23,24]. It has also been reported that *CRK5* overexpression enhances plant sensitivity to ABA, promotes stomatal closure, and improves drought resistance [25]. Our studies confirmed that CRK5 plays a predominant role in UV-induced photooxidative stress responses and suggest its involvement in the chloroplast retrograde NPQ- and ROS-dependent signaling [26]. During a reverse genetic screen of recessive mutants for the whole CRKs family, the *crk5* line showed a striking phenotype manifested by retarded growth, elevated hydrogen peroxide, and increased susceptibility to UV radiation, suggesting the role of this protein as a negative regulator of cell death [17]. However, the molecular mechanism of CRK5 activity has not been examined. Nothing is known about the ligands, which bind to its receptor domain, nor its downstream signaling pathways. In the present work, we found that CRK5 and WRKY53 transcription factors are antagonistic regulators of chlorophyll content and leaf aging, and WRKY53 upregulates CRK5 expression after binding to its promoter region. Gas exchange analysis and measurements of foliar temperature dynamics revealed that CRK5 acts as a negative regulator of stomatal conductance, suggesting its essential role in the optimization of transpiration and water-use efficiency in plants.

## 2. Results

### 2.1. CRK5 Negatively Regulates Plant Senescence

The CRK5 receptor-like kinase was described before as a negative regulator of cell death. Even young *crk5* seedlings showed early decomposition of cotyledons shortly after the development of true leaves [26]. Moreover, our observations revealed that *crk5* mutant plants showed accelerated leaf yellowing, compared to the wild type, which was reverted in complementation lines. To get a closer insight into gene expression changes during this developmental stage, we performed whole transcriptome sequencing for Col-0, *crk5*, and two complementation lines (oeCRK5_crk5_#1, oeCRK5_crk5_#2) at the age of 4 and 6 weeks, since, as shown before, during this period plants showed the first symptoms of foliar senescence manifested by the decrease in total chlorophyll content [26]. Phenotypic and genetic characterization of all genotypes used in this study was presented in Appendix A. As shown, in spite of decreased biomass production in *crk5*, all genotypes displayed similar bolting time, which suggests comparable developmental growth between genotypes at each time point used in the study. Both complementation lines showed a similar phenotype to one another, but oeCRK5_crk5_#1 displayed higher *CRK5* expression and more significant changes in transcriptional pattern in comparison to *crk5* (Figure 1 and Appendix A); thus, this line was used for further experiments presented in this paper. In 4-week-old plants, almost 300 genes were differentially expressed in *crk5* compared to the wild type (Appendix A). The transcriptional response of senescence markers was not particularly relevant at this developmental stage, with only several slightly upregulated genes (*SAG12*, *SAG13*) (Figure 1A). However, the gene ontology (GO) enrichment analysis demonstrated that GO terms related to JA signaling and biosynthesis were the most significantly overrepresented among genes induced in 4-week-old *crk5* (Appendix A). Increased JA content and the expression of its biosynthetic genes are known to play a role in ROS induction, leading to the onset of aging processes [27]. Consistent with that, the 6-week-old *crk5* plants showed visible symptoms of accelerated leaf aging, supported by significant alterations in transcriptomic profiles, with more than 2500 differentially expressed genes, compared to the wild type (Appendix A). The analysis displayed a clear upregulation of many senescence markers, such as *SAGs*, *SIRK*, *MYB2*, *NYE1*, and *ANAC029* (Figure 1A), while GO term analysis revealed overrepresentation of induced genes involved in aging-related processes, such as leaf abscission, protein repair, sulfur, and nitrogen starvation (Appendix A). Gene expression changes observed in the mutant plants result specifically from the dysfunction of CRK5 activity since they were almost completely reverted in two independent complementation lines involved in this study (Appendix A).

### 2.2. CRK5 Promoter as a Target for WRKY53

The promoter region of *CRK5* contains a large number of W-box *cis*-elements (TTGAC(C/T)), which are targets of WRKY TFs [26,28]. Thus, our RNA-seq analysis also involved the expression pattern of WRKYs, which are likely to bind to the *CRK5* promoter and regulate its activity. During the study, in 4-week-old plants, we did not find noticeable expression changes within this gene family; however, in older plants, a clear upregulation of many *WRKYs* was found in *crk5* (Figure 1B). Subsequently, we performed a yeast one-hybrid assay to check the possible interaction between the *CRK5* promoter and several WRKY TFs, which have been previously described in terms of senescence regulation. To our surprise, out of six proteins selected for this analysis (WRKY6, WRKY22, WRKY30, WRKY53, WRKY54, WRKY70), we managed to identify exclusively one specific interaction between *CRK5* promoter and WRKY53 (Figure 2A), a positive regulator of leaf aging [29,30]. Next, to verify if WRKY53 can regulate *CRK5* expression, we performed a luminescence assay based on the transient transformation of tobacco leaves. Leaves were infiltrated with *Agrobacterium* carrying two different constructs simultaneously in three combinations: (1) *35S_pro_:WRKY53* and *35S_pro_:HMGB14* (negative control without luciferase activity), (2) *35S_pro_:HMGB14* and *CRK5_pro_:LUC* (control for luciferase background), (3) *35S_pro_:WRKY53* and *CRK5_pro_:LUC*. The HMGB14 (high mobility group B protein 14, At2g34450) was selected for this study since, similarly to WRKY53; it is a nuclear-located protein possessing DNA-binding transcription factor activity [31]. The analysis clearly showed a correlation between constitutive expression of *WRKY53* in tobacco leaves and increased (2–5 times depending on the analyzed plant) luciferase activity from *CRK5*-driven promoter compared to leaf areas infiltrated with control HMG14 construct (Figure 2B and Appendix A), which, together with yeast one-hybrid results, suggests that WRKY53 binds to *CRK5* promoter and upregulates its expression.

### 2.3. Antagonistic Regulation of Chlorophyll Content by CRK5 and WRKY53

Since WRKY53 is known to play a positive regulatory role in plant aging processes, molecular evidence that this transcription factor can regulate CRK5 activity encouraged us to check whether the introduction of the *wrky53* mutation into the *crk5* background reverts its accelerated senescence phenotype. Indeed, we did not observe pronounced symptoms of leaf aging in 6-week-old *crk5 wrky53* plants under ambient light; thus, we decided to use dark-induced senescence conditions. The experimental setup was based on placing plants in continuous darkness for several days since it is known that prolonged light deprivation accelerates aging processes [32]. As expected, the *crk5* plants displayed visible symptoms of enhanced yellowing of leaves after being kept four days in constant darkness, while the *wrky53* mutant line showed a similar phenotype to wild-type plants. Interestingly, the complementation line (oeCRK5_crk5_#1) and *crk5 wrky53* double mutant managed to revert the *crk5* phenotype (Figure 3A). To further assess the dynamics of physiological changes as a result of light deprivation, we measured the level of photosynthetic pigments in the analyzed genotypes. As expected, constant darkness led to enhanced degradation of pigments (chlorophyll *a*, chlorophyll *b*, and carotenoids) in all analyzed genotypes, although the chlorophyll *a*/*b* ratio remained unchanged (Figure 3B–E). As expected, taking into account the function of WRKY53 in the promotion of leaf aging processes, the *wrky53* plants showed significantly increased foliar chlorophyll *a* and *b* levels both under ambient light and after keeping plants in continuous darkness compared to the wild-type, while the *crk5* line showed the opposite phenotype. Most intriguingly, however, the *crk5 wrky53* double mutant plants recovered the wild-type phenotype in terms of chlorophyll content; its concentration was significantly different than in both single mutants, lower compared to *wrky53* and higher compared to *crk5* (Figure 3B,C), which suggests that these two regulatory proteins play opposite roles in signaling pathways controlling chlorophyll degradation processes. The ion leakage analysis revealed an increased membrane permeability in *crk5* after keeping plants for four days in the darkness, as indicated by their nearly 3-fold increase in electrolyte outflow in comparison to the wild-type (Appendix A). Interestingly, the 4-week-old *crk5* plants used in these studies showed slightly decreased pigment concentration and increased cell death symptoms even under ambient light conditions (Figure 3B–E and Appendix A). These differences were not particularly relevant but suggested early senescence processes ongoing in this mutant line already at this developmental stage.

Later on, we performed quantitative PCR analysis to check the transcriptional response of *WRKY53* and *CRK5* in plants lacking CRK5 and WRKY53 activity, respectively. The study involved plants at different developmental stages to evaluate if the expression pattern is age-dependent. The *wrky53* line showed reduced *CRK5* expression in most of the analyzed time points. On the other hand, younger, 3-week-old plants did not show any differences in *WRKY53* expression, although we found a significant upregulation of this gene in older *crk5* plants, especially in 5- and 6-week-old, which corresponds with accelerated senescence phenotype (Figure 1 and Figure 4A). Based on these results, we created a proposed model illustrating the mutual WRKY53 and CRK5 relationship (Figure 4B).

### 2.4. The Role of CRK5 in Control of Stomatal Conductance and Foliar Temperature

The analysis of gas exchange revealed that stomatal conductance and evapotranspiration were significantly higher in 4-week-old *crk5* plants, especially under lower CO_2_ concentration, reaching almost two times higher values than in the other analyzed genotypes. These differences were even larger in 6-week-old senescing *crk5* plants, which showed even more deregulated and disturbed transpiration rates (Figure 5A,C). Consistent with this, the detached leaves from the *crk5* mutant also exhibited a faster relative water loss rate compared to the wild-type (Appendix A). At the same time, all analyzed plants did not show any differences in the CO_2_ assimilation rate (Figure 5B). To check whether increased stomatal conductance in *crk5* does not result from an increased number of stomata, the analysis of stomatal number was performed. The *crk5* plants showed slightly larger stomatal density, but the differences, compared to the wild-type, were not so large to explain the observed phenotype (Appendix A). Since enhanced transpiration rate did not result in higher CO_2_ uptake in *crk5* plants during photosynthesis (Figure 5B,C), we found significantly lower water-use efficiency in this genotype, which was reverted in *crk5 wrky53* double mutant (Figure 5D). The observation that CRK5 plays a negative regulatory role in stomatal conductance was supported by foliar temperature changes using a high-resolution thermal camera and methodology developed in our laboratory [33]. It was shown before that foliar temperature increase under growing light intensity is directly proportional to chlorophyll content and non-photochemical quenching (NPQ) increase while it is inversely proportional to stomatal aperture [33]. In this study, the *crk5* plants displayed significantly lower foliar temperature gradient under the episode of excess light exposure compared to the wild-type, while the complementation line with enhanced *CRK5* expression showed the opposite effect, suggesting that the level of *CRK5* expression has an important regulatory role in these processes (Figure 6B,C). As expected, lanolin application to the abaxial part of the plant leaf, which blocked gas exchange, bridged the differences in foliar temperature gradient between genotypes, clearly indicating that it is an indirect effect of deregulated stomatal conductance in *crk5* plants (Appendix A).

To check whether, due to its regulatory effect on stomatal conductance, CRK5 affects resilience to water deficits, we subjected the plants to osmotic stress, which is known to affect ion homeostasis, turgor maintenance, and water availability to plant cells [34]. The 3-week-old soil-grown plants were treated with 450 mM NaCl continuously for 14 days, and chlorophyll fluorescence, and the cell death rate of the plants were measured. The *crk5* line showed enhanced sensitivity to salt treatment, manifested by loss of cell membrane integrity and more pronounced inhibition of leaf growth (Figure 7A–C). It was accompanied by a more significant decrease in the maximum quantum efficiency of photosystem II (*F*_v_/*F*_m_), which has previously been shown to be more affected in plants showing increased sensitivity to salt stress [35]. In our experimental setup, *F*_v_/*F*_m_ was significantly lower in the *crk5* line four days after salt treatment (Figure 7C). As expected, the gas exchange characteristics in salt-treated plants showed even more pronounced differences between *crk5* and wild-type in terms of transpiration and stomatal conductance, supporting that this receptor-like kinase might play an important role in the regulation of stomatal aperture under osmotic stress (Appendix A). Intriguingly, the *crk5 wrky53* double mutant line did not fully restore the wild-type phenotype under salt treatment, suggesting the existence of other proteins which could be involved in this conditional negative feedback regulatory loop.

Our preliminary studies, based on yeast two-hybrid cDNA library screening, identified 32 proteins interacting with CRK5 (Appendix A). Intriguingly, as many as six of these interactors were previously described in terms of their role in response to water deprivation or calcium signaling (At5g47100, At5g17860, At3g14080, AT4G26080, AT1G67360, AT5g67480) (Appendix A) Among the other identified CRK5 interactors we also found three proteins associated with cell wall modification (AT1G48100, AT5G62360, AT4G14130), which are involved either in the composition of guard cell walls or regulation of stomatal movements in response to abiotic stress. A detailed description of these nine selected proteins is presented in Appendix A. All of them are promising candidates for participation in CRK5 regulation of stomatal conductance, but it requires further detailed studies.

## 3. Discussion

In the present study, we provided molecular evidence that CRK5 acts as a negative regulator of senescence. Transcriptomic data confirm that premature leaf aging in *crk5* plants is followed by the deregulation of many genes, especially many WRKY transcription factors. One of them, WRKY53, has been shown to bind to the *CRK5* promoter and activate its expression; however, significantly enhanced *WRKY53* expression in *crk5* mutant plants suggests the existence of a negative feedback regulatory loop between these proteins. In this study, we showed that CRK5 and WRKY53 are antagonistic regulators of stomatal conductance, which affects water-use efficiency and foliar temperature. Our previous studies also showed that plants impaired in CRK5-mediated signaling display lower biomass production rates [17,26], consistent with other reports showing a positive correlation between *CRK5* expression and leaf growth [23]. In our RNA seq study, 4-week-old plants did not show the relevant response of most senescence markers, except an increased abundance of *SAG12* and *SAG13*, while GO term analysis revealed differential expression of many genes involved in jasmonic acid (JA) biosynthesis and signaling, as well as a cellular response to wounding (Appendix A), which is accompanied by induced JA accumulation [36,37], suggesting that CRK5 might affect this hormonal signaling pathway. Accumulation of JA induces ROS, which damages chloroplasts and leads to photosynthetic dysfunction, induction of cell death, and senescence processes [38]. Therefore, our RNA-seq results show significant changes in the expression of genes associated with JA biosynthesis and signaling in 4-week-old *crk5* plants corresponding with their enhanced H_2_O_2_ content and increased activity of oxidoreductive enzymes, which was described in our previous studies [26]. It has recently been reported that increased JA biosynthesis is induced by Oxidative Stress Inducible 1 (OXI1) kinase [39]. The OXI1 overexpressing lines show a similar phenotype to *crk5* mutant plants, manifested by enhanced cell death and induced early senescence of mature leaves [40]. Our RNA-seq analysis revealed a significant upregulation of *OXI1* in 6-week-old *crk5* plants. Therefore, it would be reasonable to check whether CRK5 overlaps the OXI1 signaling pathway in the regulation of cell death and aging processes in future studies.

The CRKs are characterized by the enrichment of W-box *cis*-regulatory elements in their promoter regions [18]. A number of studies have shown that these short DNA sequences are specifically recognized by WRKY proteins, which are known to be involved in the activation of SA biosynthesis genes and the activation of systemic acquired resistance [28,41,42]. In accordance with that, the transcriptomic screen of the whole CRK family showed an SA-regulated expression pattern [18]. Our previous studies also revealed significantly elevated SA levels in *crk5* plants [26]. As expected, based on the enrichment of W-box *cis*-elements in the *CRK5* promoter, we found a significant induction of a number of WRKY TFs in 6-weekold *crk5* plants (Figure 1B). However, a yeast one-hybrid system-based study, which involved six WRKYs described in terms of senescence regulation, identified exclusively one protein, WRKY53, which directly interacted with the *CRK5* promoter (Figure 2). Opposite to CRK5 kinase, the WRKY53 is a transcription factor and acts as a positive regulator of cell death, which regulates many genes involved in transport and remobilization processes, as well as ROS-scavenging enzymes, e.g., catalases [8,29,30]. Our previous studies showed that CRK5 also affects H_2_O_2_ levels and controls the activity of ROS-scavenging enzymes, such as superoxide dismutase, catalase, or ascorbate peroxidase [26]. Quantitative PCR data revealed significantly decreased *CRK5* expression in the *wrky53* mutant line (Figure 4A), consistent with the transient transactivation assay showing a positive regulatory impact of WRKY53 on CRK5 (Figure 2B). On the other hand, we found significantly increased *WRKY53* expression in *crk5* even in 4-week-old plants, which was reverted in complementation lines (Figure 4A). It suggests that CRK5 might have a negative regulatory effect on WRKY53. However, due to the different subcellular localization of these proteins (plasma membrane vs. nucleus), this signaling pathway is likely to involve additional components, which act as secondary modulators of WRKY53. Interestingly, the early-senescence phenotype observed in *crk5* is very similar to *why1* plants, which are impaired in the activity of single-stranded DNA-binding WHIRLY1 (WHY1) and also show elevated *WRKY53* expression [43]. An intriguing feature of WHIRLY1 is its dual location in plastids and the nucleus of the same cell, which makes it a good candidate as a redox sensor in chloroplast-to-nucleus retrograde signaling [44]. Thus, it cannot be excluded that CRK5 might act upstream of WHY1 in the regulation of *WRKY53* expression and subsequent aging processes, but it needs further investigation.

In our studies, the introduction of the *wrky53* mutation reverted the *crk5* phenotype in terms of accelerated senescence, which was supported by increased permeability of the membranes (Appendix A). Most intriguingly, however, we found that the *crk5 wrky53* double mutant recovered the wild-type phenotype in terms of chlorophyll *a* and *b* content under prolonged darkness conditions, while *wrky53* and *crk5* plants displayed significantly increased and decreased levels of these pigments, respectively (Figure 3B,C). The fact that the *crk5 wrky53* double mutant showed relevant differences in chlorophyll concentration in comparison to both appropriate single mutants strongly suggests that CRK5 and WRKY53 play opposite roles in signaling pathways controlling chlorophyll degradation processes. Therefore, taking into consideration transcriptional deregulations, yeast-one-hybrid assay, and senescence phenotypes in single and double mutants, these proteins create a conditional negative feedback loop. WRKY53 binds to the CRK5 promoter and activates its expression. Subsequently, CRK5 proceeds signaling pathway leading to WRKY53 suppression (Figure 4B).

During the studies on the function of CRK5 in senescence regulation, we performed gas exchange analysis, which revealed that this receptor-like kinase has a negative regulatory effect on stomatal conductance and evapotranspiration, leading to enhanced water-use efficiency (Figure 5). Our thermal camera-based analyses of foliar temperature under variable light conditions strongly support that CRK5 regulates stomatal aperture and subsequent gas exchange since higher foliar temperature gradient under increasing light, which results from lower transpiration rate, seems to be positively correlated with *CRK5* expression level (Figure 6). In accordance with that, lanolin application to the abaxial part of the leaf, which blocked gas exchange, bridged the differences in foliar temperature during excess light stress between the analyzed lines (Appendix A). The relationship between chlorophyll content, NPQ, foliar temperature, and stomatal conductance has previously been described, and a mathematical model describing these processes was proposed [33]. It is important to note here that heat during excess light stress is generated in photosynthesis reactions centers (P680 and P700) and light-harvesting antenna (LHCII and LHCi) during NPQ and energy quenching (qE) as heat and is directly dependent on chlorophyll content and light antenna conformations [33]. Thus, heat generation in chloroplasts is a primary event, which subsequently affects stomatal conductance and transpiration changes leading to fine cooling or heating of the leaf. For better understanding, this can be compared to the nuclear plant where a nuclear reactor is a heat source like chloroplasts, while heat intercoolers and heat exchangers between primary and secondary water loops in a nuclear plant can be compared to stomatal conductance and transpiration. The observation that *crk5* plants showed lower foliar temperature under excess light stress corresponds with their lower chlorophyll content (Figure 3) and with our previous data showing significantly lower NPQ in these mutant plants [26], which indicates that less absorbed energy was converted and dissipated as heat [33,45]. It is in accordance with our [26,33,46] and other reports showing a negative correlation between stomatal conductance and NPQ [47,48].

Our findings that CRK5 acts as a negative regulator of stomatal opening in *Arabidopsis* are consistent with previous reports showing that CRK5 overexpression increases ABA sensitivity and promotes stomatal closure [35]. Our yeast two-hybrid system study revealed that CRK5 interacts with ABI1 (ABSCISIC ACID INSENSITIVE 1) phosphatase (Appendix A), which acts as a negative ABA regulator [49]. Therefore, it cannot be excluded that CRK5 might act as a ligand-binding protein that suppresses ABI1 phosphatase activity, thus activating ABA-mediated signaling pathways, but it requires further investigation. Interestingly, the *crk5 wrky53* double mutant managed to revert *crk5* phenotype in terms of disturbed stomatal conductance, transpiration, and water loss (Figure 5 and Appendix A), suggesting that CRK5 and WRKY53 could also antagonistically regulate ABA-mediated stomatal opening and closure. Previous reports also support the role of WRKY53 in plant response to water deprivation as well as in the regulation of stomatal conductance and transpiration [50]. The *WRKY53* overexpressing plants were reported to have increased tolerance to salt, while *wrky53* mutant plants accumulated higher ABA contents [15].

Due to deregulated water-use efficiency, *crk5* plants might have more problems with turgor maintenance, which could explain their increased sensitivity to osmotic stress, manifested by enhanced cell death, growth arrest, and decreased photosynthetic efficiency compared to wild-type plants (Figure 7). This conclusion is consistent with previous studies showing that overexpression of *CRK5* enhanced plant tolerance to transient water deficit [25]. Interestingly, the introduction of *wrky53* mutation into the *crk5* background has not completely reverted osmotic stress phenotype to wild-type plants (Figure 7B,C), suggesting the presence of other proteins which interact with CRK5 or are indirectly involved in its signaling in this regulatory pathway. Among 32 identified CRK5 interactors, we found six proteins involved in calcium or ABA signaling (Appendix A), likely to play a role in water deficit response. Apart from the above-mentioned ABI1, we identified several interactors associated with calcium signaling, acting either as Ca^2+^ transporters (CAX7) or possessing calcium/calmodulin-binding domains (CBL9, BT4) [51,52,53]. It has been widely described that drought or osmotic stress signals trigger a sudden increase in the cytoplasmic Ca^2+^, which subsequently affects membrane polarization, stomatal closure in guard cells, as well as phosphorylation of ion channels and aquaporins [54]. Moreover, our screen identified three other CRK5 interactors (PGX3, PMEI13, XTH15) associated with cell wall modification (Appendix A), which are likely to change the architecture of guard cell walls and influence the dynamics of stomatal opening and closure in response to drought or osmotic stress [55,56,57].

Taken together, the above data suggest that receptor-like kinase CRK5 and WRKY53 transcription factors are conditional and antagonistic regulators of leaf aging and water-use efficiency, as well as transpiration and foliar temperature. CRK5 might also regulate ABA- and calcium-dependent-signaling pathways during water deficit, but unraveling which of these newly identified CRK5 interactors play a predominant role in these processes requires further research.

## 4. Materials and Methods

### 4.1. Plant Material and Growing Conditions

All *Arabidopsis thaliana* plants used in this research were in the Col-0 background. The T-DNA insertional mutant seeds of *crk5* (SALK_063519C) and *wrky53* (SALK_034157C) were ordered from the Nottingham Arabidopsis Stock Center (NASC) and checked by PCR. The generation of complementation lines, and the level of *CRK5* expression in *crk5* and complementation lines were previously described [26]. In complementation lines, the cDNA of *CRK5* was driven by the 35S promoter. The *crk5 wrky53* double mutants were generated by crossing. All primers used for genotyping are shown in Appendix A.

After stratification (3 days at 4 °C), plants were placed in the growing chamber and grown on Jiffy Pots for 4–6 weeks in short-day photoperiod (8 h/16 h) under the constant white light of 120 µmol photons m^−2^ sec^−1^ at 22 °C, relative humidity of 70 ± 5%. Short-day photoperiod was used to delay plant vegetative to reproductive phase transition in order to exclude transcriptional changes during bolting in senescence-related experiments. For dark-induced senescence studies, 4-week-old plants were kept in continuous darkness for four days.

### 4.2. Vector Constructions and Plant Transformation

For luciferase studies, the coding sequences of *WRKY53* and *HMG*, as well as the 1248 bp promoter region of *CRK5*, were amplified with primers listed in Appendix A. PCR products were purified and cloned into the entry (pENTR/D-TOPO) clone. Next, the *WRKY53* and *HMG* CDSs were subcloned into pGWB641, while the *CRK5* promoter was subcloned into the pGWB635 vector in front of a luciferase reporter gene. These constructs were inserted into *Agrobacterium* strain GV3101 and used for the transient transformation of *Nicotiana benthamiana* leaves.

### 4.3. Real-Time PCR

For the age-related gene expression study, RNA was isolated from the seventh leaves of plants at different ages (from 3 to 7 weeks old). RNA for qRT-PCR was extracted as described previously [58]. UPL7 and PP2AA3 were used as reference genes to calculate relative expression. All primers used in this study are listed in Appendix A.

### 4.4. RNA Sequencing

For RNA isolation, seventh leaves from 4- or 6-week-old plants were used. The experiment was performed as described before [59] in three biological replicates. Each replicate consisted of several pooled detached leaves from at least three individual plants. Significantly enriched GO terms were identified in down-/upregulated gene sets using the PANTHER Classification System.

### 4.5. Availability of Supporting Data

RNA-seq data are provided at (uploaded upon manuscript acceptance).

### 4.6. Transient Transactivation Assay

The study was based on the transient transformation of *N. benthamiana* leaves with two genetic constructs simultaneously: (1) *35S_pro_:WRKY53* and *35S_pro_:HMGB14* (negative control without luciferase activity), (2) *35S_pro_:HMGB14* and *CRK5_pro_:LUC* (control for luciferase background), (3) *35S_pro_:WRKY53* and *CRK5_pro_:LUC*. Luciferase activity was quantified in 12 independent tobacco plants. Collected samples (~5 mg) were ground in 0.5 mL lysis buffer (Promega kit). 50 μL of the homogenate was placed under a luminometer tube (Berthold), and 50 μL of luciferin assay was added 10 s before the measurement. RLUs were expressed per gram fresh weight of leaves.

### 4.7. Yeast One-Hybrid System

The analysis was performed using Matchmaker Gold Yeast One-Hybrid System (Clontech Laboratories, Mountain View, CA, United States). The coding sequences of *WRKY6*, *WRKY22*, *WRKY30*, *WRKY53*, *WRKY54*, and *WRKY70*, as well as the promoter sequence of *CRK5* (1248 bp length), were amplified and cloned using the pENTR/D-TOPO vector (primers used for amplification were shown in Appendix A). Then, the WRKYs (prey) constructs were digested by NdeI/EcoRI and BamHI restriction enzymes and ligated into the pGADT7 vector, while the construct containing *CRK5* promoter was digested by KpnI and XhoI and inserted into pAbAi to generate bait vector pCRK5-AbAi. The pCRK5-AbAi and p53-AbAi (control) plasmids were then integrated into the Y1HGold yeast genome. All constructed WRKY-pGADT7 vectors, together with empty pGADT7 (control), were then transformed into Y1HGold strain containing either pCRK5-AbAi or p53-AbAi. From each of the transformation reactions, 100µl of 1/100 dilution was spread on SD/-Leu plates (transformation control) and SD/-Leu with Aureobasidin A (200 ng/mL). Colonies were grown for four days at 30 °C. Afterward, several colonies were picked and analyzed by colony PCR using the Matchmaker Insert Check PCR Mix.

### 4.8. Screening for CRK5 Interactors by Yeast Two-Hybrid Assay

The yeast two-hybrid screening assays were based on Matchmaker Gold Yeast Two-hybrid System. For the identification of CRK5 interacting proteins, Mate & Plate™ Library—Universal Arabidopsis (Normalized) (Takara Bio Company) was used. The cDNA library was introduced into the prey vector pGADT7 (AD), and the *CRK5* coding sequence was inserted in the bait vector pGBKT7 (BD). The vectors harboring the cDNA library and *CRK5* were transformed into the yeast cells Y2H Gold and Y187, respectively, which were examined using double-deficiency (SD/-Leu/-Trp), and quadruple-deficiency (SD/-Leu/-Trp/-Ade/-His with 200 ng/mL Aureobasidin) screening assays. Positive colonies grown for four days at 30 °C on the quadruple-deficiency screening plates were selected as templates, and PCR amplification and sequencing were conducted to detect the inserted fragments. The identification of interacting proteins was made using the NCBI database.

### 4.9. Gas Exchange Analysis

Gas exchange parameters were measured using a portable gas-exchange system CIRAS-3 (PP Systems, Amesbury, MA, USA) in variable conditions of CO_2_ concentration. During measurements, the CO_2_ concentration in the leaf cuvette was maintained at different levels (according to Ci Ramp protocol), humidity at 60%, light intensity at 200 μmol m^−2^ s^−1^, and temperature at 25 °C. Stomatal conductance (gs), transpiration (T), and net CO_2_ assimilation rate (Pn) were measured simultaneously. Intrinsic water use efficiency (iWUE) was calculated as the ratio of Pn to transpiration rate T: iWUE = Pn/T.

Determination of foliar temperature under variable light conditions

The analysis was performed on whole soil-grown *Arabidopsis* rosettes using a FLIR A600-Series thermal imaging infrared camera. Measurement was based on the authorial program in the labview environment. We applied variable light conditions as follows: 30 s of ambient light (150 μmol m^−2^ s^−1^), followed by 60 s of high blue light (4000 μmol m^−2^ s^−1^), followed by 60 s of ambient light (150 μmol m^−2^ s^−1^), which served as recovery stage. The background temperature during this experiment increased by about 2 °C. Analysis of thermograms was performed using OriginPro 8 software.

### 4.10. Photosynthetic Pigment Analysis

The analysis of chlorophyll *a*, chlorophyll *b*, and carotenoid contents was performed using the Multiskan GO spectrophotometer, as described previously [59]. The level of pigments was calculated according to equations reported previously [60].

### 4.11. Water Loss

For the analysis of transpiration rates, 4-week-old rosettes were cut using a scalpel and weighed immediately. Then, the leaves were placed under the light density of about 120 μmol m^−2^ s^−1^, with a relative humidity of about 70%, and weighed again after 1.5, and 3 h. The experiment was performed with 20 biological repeats for each genotype. The relative water loss rate was calculated as the ratio of water loss at each time point to the initial fresh weight.

### 4.12. Chlorophyll a Fluorescence and Growth Dynamics

The dynamics of growth (measured as differences in whole rosette area) and maximum quantum efficiency of photosystem II (PSII) under osmotic stress treatment were determined with the use of FluorCam 800 MF (Photon Systems Instruments) as described before [26].

### 4.13. Calculation of Stomatal Density

To calculate the stomatal density (defined as the number of stomata per mm^2^), abaxial epidermal strips from similarly developed leaves were analyzed with the use of a confocal microscope. The number of stomata was calculated with the use of Quick PHOTO Micro 3.1 software. Mean values [±standard deviation (SD)] were derived from the leaves of eight different plants. For each leaf, stomata were counted from three randomly chosen 0.312 mm^2^ picture areas.

### 4.14. Relative Electrolyte Leakage

The leaves were excised and transferred into 50 mL falcon tubes containing 35 mL Milli-Q water. The relative electrolyte leakage was measured with a conductance meter (WTW, INOLAB Cond Level 1) and calculated as a ratio between the value obtained after 1 h incubation and the total leakage evaluated after autoclaving the samples.

### 4.15. Statistical Analysis

The statistical analysis of Real-Time PCR was made in the “R” programming environment version 3.3.3 using stats packages. Statistical analyses of chlorophyll *a* fluorescence, gas exchange, pigments content, relative electrolyte leakage, and water loss were performed using Statistica 13 software.

### 4.16. Accession Numbers

The following gene names were used in Figure 1: HAI1 (AT5G59220), ERD1 (AT5G51070), BIR1 (AT5G48380), SAG12 (AT5G45890), ORE1 (AT5G39610), SAG14 (AT5G20230), SAG101 (AT5G14930), WRKY53 (AT4G23810), NYE1 (AT4G22920), NYC1 (AT4G13250), NYE2 (AT4G11910), SAG21 (AT4G02380), ANAC053 (AT3G10500), MYB2 (AT2G47190), JUB1 (AT2G43000), SAG13 (AT2G29350), SIRK (AT2G19190), ANAC029 (AT1G69490), WRKY6 (AT1G62300), BFN1 (AT1G11190), WRKY51 (AT5G64810), WRKY8 (AT5G46350), WRKY50 (AT5G26170), WRKY30 (AT5G24110), WRKY72 (AT5G15130), WRKY75 (AT5G13080), WRKY31 (AT4G22070), WRKY41 (AT4G11070), WRKY47 (AT4G01720), WRKY45 (AT3G01970), WRKY46 (AT2G46400), WRKY55 (AT2G40740), WRKY33 (AT2G38470), WRKY60 (AT2G25000), WRKY15 (AT2G23320), WRKY59 (AT2G21900), WRKY40 (AT1G80840), WRKY36 (AT1G69810), WRKY63 (AT1G66600), WRKY71 (AT1G29860) and WRKY61 (AT1G18860).

The following gene names were used in Figure 2: HMGB14 (AT2G34450), WRKY6 (AT1G62300), WRKY22 (AT4G01250), WRKY30 (AT5G24110), WRKY53 (AT4G23810), WRKY54 (AT2G40750), WRKY70 (AT3G56400).

The following gene names were used in Appendix A: CBL9 (AT5G47100), CAX7 (AT5G17860), BT4 (AT5G67480), ABI1 (AT4G26080), LDAP1 (AT1G67360), LSM1B (AT3G14080).

## Figures and Tables

**Figure 1 cells-11-03558-f001:**
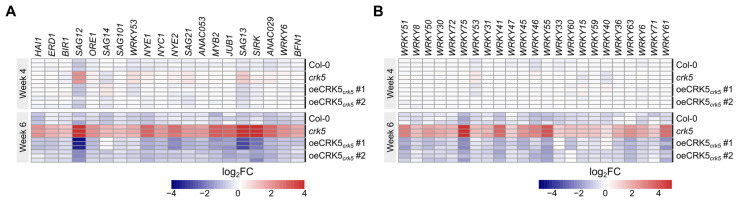
RNA seq analysis of plants at different ages (four and six weeks old). The study involved four genotypes: Col-0, *crk5* and two complementation lines oeCRK5crk5#1, oeCRK5crk5#2. Heat map shows the expression patterns of selected senescence marker genes (**A**) and WRKY transcription factors (**B**).

**Figure 2 cells-11-03558-f002:**
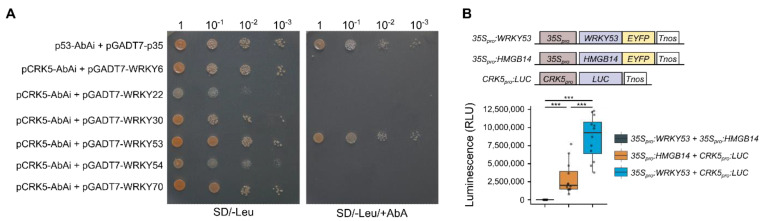
(**A**) Identification of interaction between *CRK5* promoter and six WRKY senescence-related transcription factors (WRKY6, WRKY22, WRKY30, WRKY53, WRKY54, WRKY70). The analysis was performed by Matchmaker Gold Yeast One-Hybrid System. Single-deficiency (SD/–Leu) screening assay was performed to estimate yeast transformation efficiency, while the addition of 200 ng/mL aureobasidin allowed for identifying potential binding partners. The combination of p53-AbAi and pGADT7-p53 was used as a positive control. (**B**) Transient transactivation luciferase reporter assay. The study was based on the transient transformation of *N. benthamiana* leaves with two genetic constructs simultaneously: (1) prom35S-WRKY53 and prom35S-HMG At2g34450 (negative control without luciferase activity), (2) prom35S-HMG promCRK5-LUC, (3) prom35S-WRKY53 and promCRK5-LUC. Mean values of luciferase activity (RLU) were derived from 12 tobacco plants (*n* = 12). Dots represent individual observations. Asterisks indicate a significant difference according to the *t*-test at level *p* < 0.001 (***).

**Figure 3 cells-11-03558-f003:**
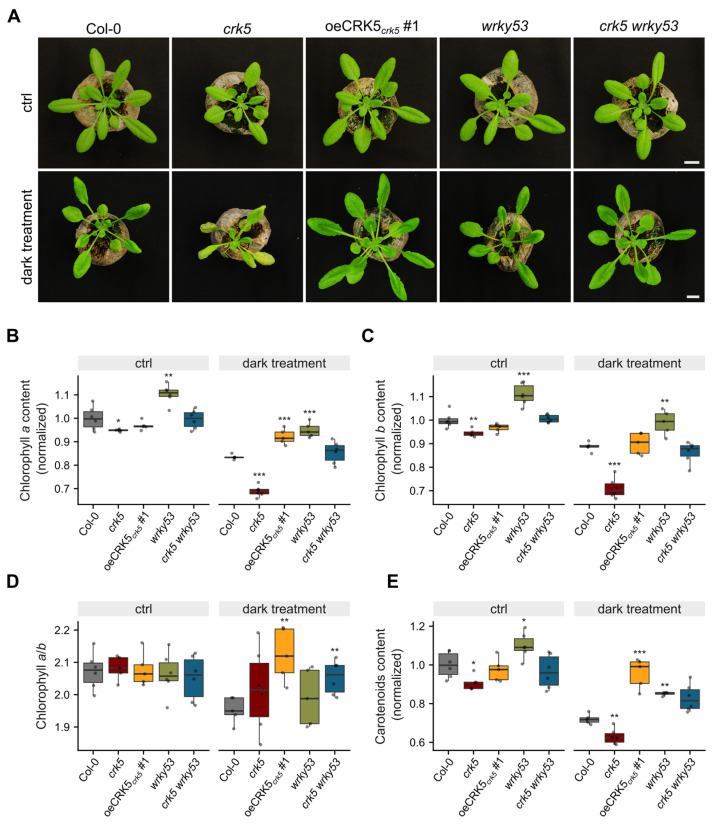
(**A**) Morphological phenotype of dark-induced senescence in the analyzed genotypes. Photographs were taken of plants growing under ambient light (120 μmol m^−2^ s^−1^) conditions and then transferred for four days to continuous darkness. White scale bars indicate 1 cm (**B**–**E**) Analysis of pigment content changes under continuous darkness conditions. The figures show the level of chlorophyll *a* (**B**), chlorophyll *b* (**C**), chlorophyll *a*/*b* ratio (**D**), and carotenoids (**E**). Mean values (±SD) were derived from 6 plants (*n* = 6). Dots represent individual observations. Asterisks indicate a significant difference according to the *t*-test at level *p* < 0.001 (***), *p* < 0.01 (**), and *p* < 0.05 (*).

**Figure 4 cells-11-03558-f004:**
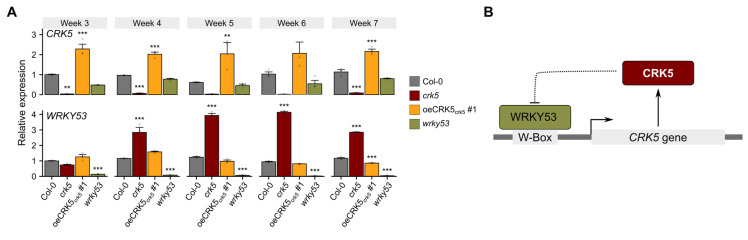
(**A**) qRT-PCR analysis of expression of *WRKY53* and *CRK5* in the analyzed genotypes. Data show relative expression normalized to the wild type and represent average values ± SD. Asterisks indicate a significant difference according to the *t*-test at level *p* < 0.001 (***), *p* < 0.01 (**). (**B**) Proposed model describing mutual CRK5 and WRKY53 relationship. We suggest the existence of a negative feedback loop between these proteins.

**Figure 5 cells-11-03558-f005:**
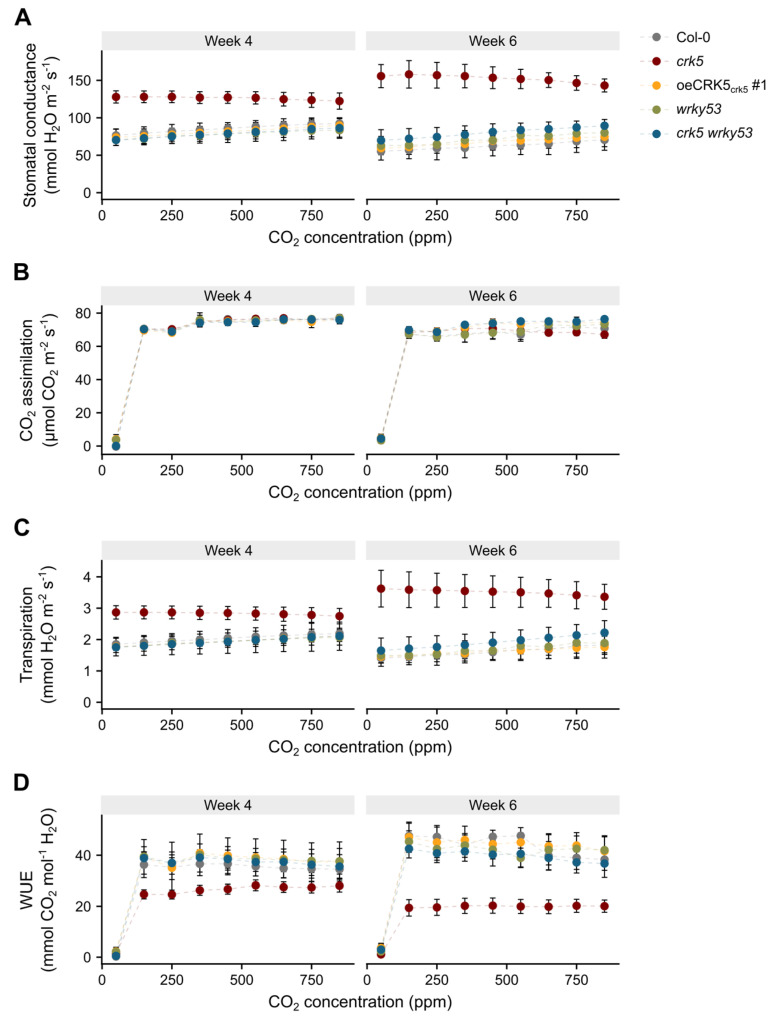
Foliar gas exchange characteristics in variable intercellular CO_2_ concentration. The analysis involved 4-week-old and 6-week-old plants and was performed on CIRAS-3 Portable Photosynthesis System, with PAR of 300 µmol m^−2^ s^−1^ and Cuvette Flow of 300 mL/min (A/Ci C3 ramp program). Individual charts represent (**A**) stomatal conductance, (**B**) CO_2_ assimilation, (**C**) evapotranspiration, and (**D**) water use efficiency (WUE). Mean values (±SD) were derived from 8 plants (*n* = 8).

**Figure 6 cells-11-03558-f006:**
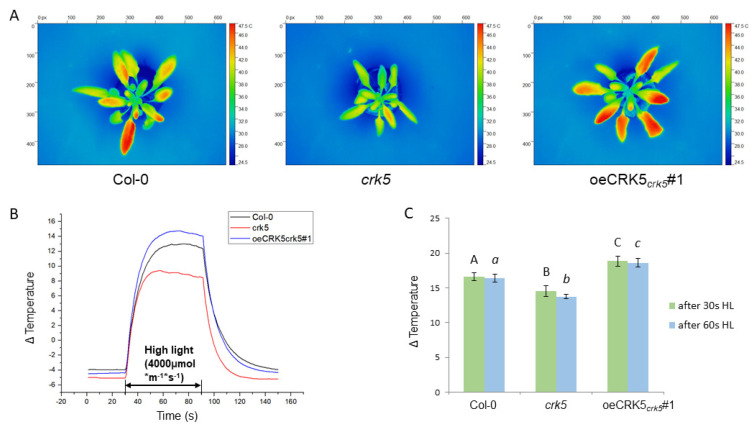
Analysis of foliar temperature under variable light conditions. (**A**) Thermograms showing whole rosettes of different *Arabidopsis* genotypes after 30 s high light (4000 μmol photons m^−2^ s^−1^). (**B**) Plot showing dynamic foliar temperature changes under variable light conditions in comparison to the background—white surface on which plants were placed during the measurement. The white surface reflected the light, and its temperature increased by only 1,5C during high light exposure. The program was set as follows: 30 s of ambient light (150 μmol photons m^−2^ s^−1^), followed by 60 s of high blue light (4000 μmol photons m^−2^ s^−1^), followed by 60 s of ambient light (150 μmol photons m^−2^ s^−1^). (**C**) Average leaf temperature changes after 30 s and 60 s of high light exposure. Data represent mean values of 4 different leaves from 5 plants (*n* = 20). Statistical analysis was performed according to a *t*-test at level *p* < 0.05. Letters A, B, C and a, b, c above the bars indicate homogenous groups.

**Figure 7 cells-11-03558-f007:**
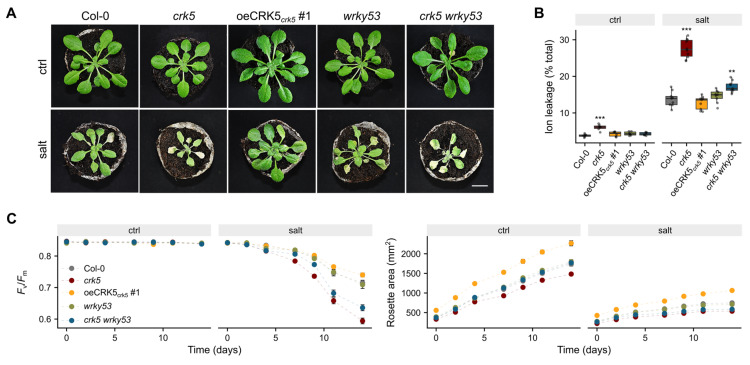
(**A**) Morphological phenotype of plants treated with 450 mM NaCl for 14 days. (**B**) Relative cellular electrolyte leakage was measured after 14 days of salt treatment compared to non-treated plants. Dots represent individual observations. Mean values (±SD) are derived from 10 plants (*n* = 10). Asterisks indicate a significant difference relative to Col-0 according to the one-way ANOVA and Tukey HSD test at level *p* < 0.001 (***), *p* < 0.01 (**). (**C**) The dynamics of maximum quantum yield of photosystem II (*Fv/Fm*) and plant growth within the osmotic stress treatment. Data represent mean values of 10 plants (*n* = 10). The statistics for Figure 7C (*p*-values of Tukey HSD test compared to Col-0) were presented in Appendix A.

## Data Availability

The data presented in this study are available on request from the corresponding author.

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
