# Peer review of "The CRK5 and WRKY53 Are Conditional Regulators of Senescence and Stomatal Conductance in Arabidopsis"

_cells, 2022, doi:10.3390/cells11223558_

Round 1
Reviewer 1 Report
General comments
This manuscript presents a well-conducted study on the functions of a transcript factor, WRKY53, and a receptor, CRK5, by using molecular and physiological approaches. Overall, the methodological strategy is scientifically sound. However, the authors should work at crafting their writing and thus make the story clearer for the readers. They should start by defining direct and indirect responses to a lack of TFs functions. For instance, the authors claim: "Obtained results suggest that WRKY53 and CRK5 are conditional regulators of senescence, chlorophyll synthesis/degradation, and stomatal conductance." But shortly after they affirmed: "We found that receptor-like kinase CRK5 and WRKY53 transcription factor are conditional and antagonistic regulators of leaf senescence, transpiration, and affect water-use efficiency and foliar temperature”. It is clear to me that leaf temperature, transpiration and water use efficiency were indirect responses to the lack of function of CRK5 and WRKY53, and a direct consequence of the decreased stomatal conductance. So, it is suitable that the authors work on a better organization of results presentation and discussion in a logical way.
Detailed comments
Abstract
Line 14: “Our studies”? or “Our study”?
Line 15: That’s a little bit confusing. You need to avoid specific explanations.
Suggestion: “Our study revealed that one member of this family, CRK5, acts as a negative regulator of leaf aging. Enrichment of CRK5 promoter region in W-box cis-elements demonstrated that it is regulated by WRKY transcription factors. Moreover, we observed significantly enhanced WRKY53 expression in crk5 and reversion of its early-senescence phenotype in crk5 wrky53 line, which suggests the existence of a negative feedback loop between these proteins, which antagonistically regulate chlorophyll a and b contents. Yeast-two hybrid assay showed further that CRK5 interacts with several proteins involved in response to water deprivation or calcium signaling, while gas exchange analysis revealed a positive effect of CRK5 on water use efficiency. Consistent with that, the crk5 plants showed disturbed stomatal conductance and increased susceptibility to osmotic stress. These traits were partially reverted to wild-type phenotype in crk5 wrky53 double mutant. In addition, the crk5 plants showed lower foliar temperatures during excess light stress. Obtained results suggest that WRKY53 and CRK5 are conditional regulators of senescence, chlorophyll synthesis/degradation and stomatal conductance.”
Introduction
The authors should be more specific about what they are attempting to investigate. Some topics are presented repeatedly without linking with study goals.
Discussion
The discussion is also long and dense and hard work to read. Please rework it by summarizing the main findings in view of the research questions in the first paragraph of Discussion. After this, authors can explore in subsequent paragraphs different aspects of the work and explain how their findings expand the envelope of knowledge; but, first of all, authors simply need to state the main results without discussing their why and how or the relationships to the literature.
Author Response
Dear Reviewer,
Please find below our answers to your comments concerning the manuscript entitled "The CRK5 and WRKY53 are conditional regulators of senescence and stomatal conductance in Arabidopsis” by Paweł Burdiak, Jakub Mielecki, Piotr Gawroński and Stanisław Karpiński".
- Reviewer 1
General comments
This manuscript presents a well-conducted study on the functions of a transcript factor, WRKY53, and a receptor, CRK5, by using molecular and physiological approaches. Overall, the methodological strategy is scientifically sound. However, the authors should work at crafting their writing and thus make the story clearer for the readers. They should start by defining direct and indirect responses to a lack of TFs functions. For instance, the authors claim: "Obtained results suggest that WRKY53 and CRK5 are conditional regulators of senescence, chlorophyll synthesis/degradation, and stomatal conductance." But shortly after they affirmed: "We found that receptor-like kinase CRK5 and WRKY53 transcription factor are conditional and antagonistic regulators of leaf senescence, transpiration, and affect water-use efficiency and foliar temperature”. It is clear to me that leaf temperature, transpiration and water use efficiency were indirect responses to the lack of function of CRK5 and WRKY53, and a direct consequence of the decreased stomatal conductance. So, it is suitable that the authors work on a better organization of results presentation and discussion in a logical way.
Our answer: Thank you for your kind comment concerning a general methodological strategy used to reveal the function and molecular relationship between CRK5 and WRKY53. Anyway, after careful critical rereading of the manuscript we fully agree that the whole story was written in quite complicated way, with repetitions and topics not directly related to our research, Therefore, we made corrections to shorten the text particularly in Introduction and Discussion. We removed some parts of the manuscript, which we found confusing for the reader.
As for your comment concerning direct and indirect consequences of CRK5 and WRKY53 disfunction, we believe that heat generation in chloroplasts due to NPQ and qE (energy quenching) is likely to be a primary event in rapid foliar temperature increase. which subsequently affects stomatal conductance, transpiration changes, leading to fine cooling or heating of the leaf. It was shown before that foliar temperature increase under growing light intensity is directly proportional to chlorophyll content and non-photochemical quenching (NPQ) increase while it is inversely proportional to stomatal aperture [33, Kulasek et al., 2016]. In the previous paper we showed that CRK5 is a positive regulator of NPQ [26, Burdiak et al., 2015]. Heat generation that influences foliar temperature is a quantum-molecular mechanism, therefore we believe it precedes physiological and biochemical regulatory responses like stomatal aperture or water-use efficiency. Therefore, we added and described this hypothesis in the Discussion.
- Reviewer 1
Detailed comments
Abstract
Line 14: “Our studies”? or “Our study”?
Our answer: of course, it should be “Our study”. It was corrected.
- Reviewer 1
Line 15: That’s a little bit confusing. You need to avoid specific explanations.
Suggestion: “Our study revealed that one member of this family, CRK5, acts as a negative regulator of leaf aging. Enrichment of CRK5 promoter region in W-box cis-elements demonstrated that it is regulated by WRKY transcription factors. Moreover, we observed significantly enhanced WRKY53 expression in crk5 and reversion of its early-senescence phenotype in crk5 wrky53 line, which suggests the existence of a negative feedback loop between these proteins, which antagonistically regulate chlorophyll a and b contents. Yeast-two hybrid assay showed further that CRK5 interacts with several proteins involved in response to water deprivation or calcium signaling, while gas exchange analysis revealed a positive effect of CRK5 on water use efficiency. Consistent with that, the crk5 plants showed disturbed stomatal conductance and increased susceptibility to osmotic stress. These traits were partially reverted to wild-type phenotype in crk5 wrky53 double mutant. In addition, the crk5 plants showed lower foliar temperatures during excess light stress. Obtained results suggest that WRKY53 and CRK5 are conditional regulators of senescence, chlorophyll synthesis/degradation and stomatal conductance.”
Our answer: Thank you for this comment. It was changed according to your suggestion. Indeed, abstract should contain only general information and specific explanations should be placed later in the presentation of results. We will remember to follow your suggestion also in future papers.
- Reviewer 1
Introduction
The authors should be more specific about what they are attempting to investigate. Some topics are presented repeatedly without linking with study goals.
Our answer: The Introduction has been modified to address your remark. In the end of Introduction we added some text to explain the study goals.
- Reviewer 1
Discussion
The discussion is also long and dense and hard work to read. Please rework it by summarizing the main findings in view of the research questions in the first paragraph of Discussion. After this, authors can explore in subsequent paragraphs different aspects of the work and explain how their findings expand the envelope of knowledge; but, first of all, authors simply need to state the main results without discussing their why and how or the relationships to the literature.
Our answer: we fully agree that the previous version of discussion was too long and the summary our main findings at the beginning was missing. Therefore, we made many corrections to Discussion. First of all, we added the summary of main results at the beginning. We also removed some parts which were already mentioned in Introduction to avoid repetitions. Finally, we removed some parts of discussion not directly related to our research to make it more interesting and accessible for the reader in order to highlight the major findings.
We hope that this corrected version of the manuscript will meet your expectations.
Yours sincerely,
Stanislaw Karpinski (on behalf of all the authors).
Reviewer 2 Report
The reviewed article presents interesting results showing that “receptor-like kinase CRK5 and WRKY53 transcription factors are conditional and antagonistic regulators of leaf aging and water-use efficiency as well as transpiration and foliar temperature”. Experiments are well done and mostly adequately described and discussed. However, I have comments concerning description of some of the methods and results and their interpretation.
1. I think it should be specified, from which plants organs RNA was isolated for PCR. I suspect that the use of Jiffy Pots did not allow root sampling and RNA was isolated from shoots. It is important to mention this, since transcriptomic response is likely to be organ specific
2. I suspect that something is wrong with the data in Figure 6b. It shows that delta temperature was lower in crk5 mutant. But according to the heat map of Figure 6, leaves of crk5 were cooler, which is likely to result from higher stomatal conductance and leaf cooling by increased transpiration. In accordance with this the difference between leaf temperature and air should be HIGHER in and not lower in crk5 mutant. It is mentioned in the discussion that “crk5 plants showed lower foliar temperature”.
3. There are a lot of repetitions in the text. The same statements may be found in both Introduction and Discussion. I recommend to start Discussion from the own results of authors obtained in the present research and then to relate them to the literature data trying to avoid repetitions of what was already mentioned in the introduction.
4. I am not sure that the authors should mention cell death so often and definitely. What they actually measured was electrolyte leakage and this parameter increases not only with the cell death, but with increased permeability of the membranes of living cells mainly related to K+ efflux from plant cells, which is mediated by plasma membrane cation conductances.
5. Line 604. “gas exchange analysis… revealed disturbed water homeostasis in crk5 plants” – but authors did not measure “water homeostasis” (neither water potential, nor relative water content). I strongly recommend authors to pay attention to these indicators in their future work. Increased stomatal conductance does not disturb leaf hydration when it is accompanied by increased tissue hydraulic conductivity enabling increased water flow to the shoots and water balance. This happens each day, when light induced stomatal opening is accompanied by increased hydraulic conductivity. At present I recommend to avoid mentioning water homeostasis in this article.
6. The same remark concerns “water deficit signaling”. It was neither measured nor detected in the present work.
Author Response
Dear Reviewer,
Please find below our answers to your comments concerning the manuscript entitled "The CRK5 and WRKY53 are conditional regulators of senescence and stomatal conductance in Arabidopsis” by Paweł Burdiak, Jakub Mielecki, Piotr Gawroński and Stanisław Karpiński".
- Reviewer 2
Comments and Suggestions for Authors
The reviewed article presents interesting results showing that “receptor-like kinase CRK5 and WRKY53 transcription factors are conditional and antagonistic regulators of leaf aging and water-use efficiency as well as transpiration and foliar temperature”. Experiments are well done and mostly adequately described and discussed. However, I have comments concerning description of some of the methods and results and their interpretation.
- I think it should be specified, from which plants organs RNA was isolated for PCR. I suspect that the use of Jiffy Pots did not allow root sampling and RNA was isolated from shoots. It is important to mention this, since transcriptomic response is likely to be organ specific
Our answer: You are right. This information was missing and indeed it is essential to provide this information due to organ specificity of expression pattern. RNA was isolated from seventh leaf of plants at specific age (from 3 to 7-week old). Taking the same leaf was essential for us in this experimental setup because we wanted to estimate CRK5 and WRKY53 expression level at specific developmental stage.
- Reviewer 2
- I suspect that something is wrong with the data in Figure 6b. It shows that delta temperature was lower in crk5 mutant. But according to the heat map of Figure 6, leaves of crk5 were cooler, which is likely to result from higher stomatal conductance and leaf cooling by increased transpiration. In accordance with this the difference between leaf temperature and air should be HIGHER in and not lower in crk5 mutant. It is mentioned in the discussion that “crk5 plants showed lower foliar temperature”.
Our answer: You are right that in our experiment crk5 leaves were cooler after high light (as shown in Figure 6a), which is consistent with higher stomatal conductance and subsequent more efficient transpiration in the mutant plants. It was not mentioned before but in Figure 6b we did not measure the difference between the temperature of the leaf and the air. Our background was the white surface, on which the plants were placed during the measurement. Because it was white, it reflected the light and its temperature did not change considerably during the measurement. It changed only by approximately 1,5C. Therefore, we can assume that our background had almost constant temperature during the experimental setup. According to this delta temperature shown in Figure 6b (referred to as difference in foliar temperature minus background temperature) was lower in crk5 plants because due to higher transpiration rate their foliar temperature increase was lower than in other genotypes. We added some additional description to Figure 6b to clarify these doubts.
It was shown before that foliar temperature increase under growing light intensity is directly proportional to chlorophyll content and non-photochemical quenching (NPQ) increase while it is inversely proportional to stomatal aperture [33, Kulasek et al., 2016]. In the previous paper we showed that CRK5 is a positive regulator of NPQ [26, Burdiak et al., 2015]. Heat generation that influences foliar temperature is a quantum-molecular mechanism, therefore we believe it precedes physiological and biochemical regulatory responses like stomatal aperture or water-use efficiency. Therefore, we added and described this hypothesis in the Discussion.
- Reviewer 2
- There are a lot of repetitions in the text. The same statements may be found in both Introduction and Discussion. I recommend to start Discussion from the own results of authors obtained in the present research and then to relate them to the literature data trying to avoid repetitions of what was already mentioned in the introduction.
Our answer: After careful, critical rereading of the manuscript we fully agree with your remark considering many repetitions in both Introduction and Discussion. Therefore, we made significant modifications to these parts of the text and removed these fragments which appear repeatedly. As a result Introduction and Discussion now are a little bit shorter, more clear and, we believe, easier to catch the major points of the text. According to your suggestions we added the summary of our main results at the beginning of discussion and afterwards we started to relate these results to literature.
- Reviewer 2
- I am not sure that the authors should mention cell death so often and definitely. What they actually measured was electrolyte leakage and this parameter increases not only with the cell death, but with increased permeability of the membranes of living cells mainly related to K+ efflux from plant cells, which is mediated by plasma membrane cation conductances.
Our answer: After your remark we paid more attention to these parts of the text and found out that we really mentioned “cell death” too often. In our previous CRK manuscript (Burdiak et al., 2015) we made more analyses on cell death (e.g. trypan blue staining), thus we found that CRK5 acts as a negative regulator of cell death. In this manuscript “cell death” appears mostly in relation to ion leakage experiments. And you are right that this measurement is an indicator of membrane stability, which not necessarily is a result of cell death but also other physiological changes. Thus, according to your suggestion, we replaced in most parts of the text “cell death” for “increased membrane permeability”. And in some other parts we simply removed this fragment because it appeared repeatedly.
- Reviewer 2
- Line 604. “gas exchange analysis… revealed disturbed water homeostasis in crk5 plants” – but authors did not measure “water homeostasis” (neither water potential, nor relative water content). I strongly recommend authors to pay attention to these indicators in their future work. Increased stomatal conductance does not disturb leaf hydration when it is accompanied by increased tissue hydraulic conductivity enabling increased water flow to the shoots and water balance. This happens each day, when light induced stomatal opening is accompanied by increased hydraulic conductivity. At present I recommend to avoid mentioning water homeostasis in this article.
Our answer: Of course, you are right that we should not use this term, since we did not make any measurements related to this. This term was removed. Anyway, thanks to your comment, we think it’s a good idea to start doing this kind of experiments in future works.
- Reviewer 2
- The same remark concerns “water deficit signaling”. It was neither measured nor detected in the present work.
Again, we fully agree that this term also shoudn’t appear in the text. It was removed.
We hope that this corrected version of the manuscript will meet your expectations.
Yours sincerely,
Stanislaw Karpinski (on behalf of all the authors).
Round 2
Reviewer 1 Report
The ms has been heavily improved and seems suitable for publication in the present form.